# A Pan-Cancer Atlas of Differentially Interacting Hallmarks of Cancer Proteins

**DOI:** 10.3390/jpm12111919

**Published:** 2022-11-17

**Authors:** Medi Kori, Gullu Elif Ozdemir, Kazim Yalcin Arga, Raghu Sinha

**Affiliations:** 1Department of Bioengineering, Marmara University, Istanbul 34854, Turkey; 2Genetic and Metabolic Diseases Research and Investigation Center, Marmara University, Istanbul 34854, Turkey; 3Department of Biochemistry and Molecular Biology, Penn State College of Medicine, Hershey, PA 17033, USA

**Keywords:** hallmarks of cancer, differential interactome, system biomarkers, druggability, personalized treatments

## Abstract

Cancer hallmark genes and proteins orchestrate and drive carcinogenesis to a large extent, therefore, it is important to study these features in different cancer types to understand the process of tumorigenesis and discover measurable indicators. We performed a pan-cancer analysis to map differentially interacting hallmarks of cancer proteins (DIHCP). The TCGA transcriptome data associated with 12 common cancers were analyzed and the differential interactome algorithm was applied to determine DIHCPs and DIHCP-centric modules (i.e., DIHCPs and their interacting partners) that exhibit significant changes in their interaction patterns between the tumor and control phenotypes. The diagnostic and prognostic capabilities of the identified modules were assessed to determine the ability of the modules to function as system biomarkers. In addition, the druggability of the prognostic and diagnostic DIHCPs was investigated. As a result, we found a total of 30 DIHCP-centric modules that showed high diagnostic or prognostic performance in any of the 12 cancer types. Furthermore, from the 16 DIHCP-centric modules examined, 29% of these were druggable. Our study presents candidate systems’ biomarkers that may be valuable for understanding the process of tumorigenesis and improving personalized treatment strategies for various cancers, with a focus on their ten hallmark characteristics.

## 1. Introduction

Cancer is one of the leading causes of death in almost every country in the world. According to GLOBOCAN, there were about 19.3 million new cancer cases and 10 million cancer deaths in 2020 [1], and these numbers make the fateful picture clear. Cancer is an uncontrolled cell-division process that leads to cell transformation through the occurrence of various and successive genetic alterations. This transformation is the result of a complex process, and this complexity makes the disease an enigma shrouded in mystery and incurable [2].

In 2000, Hanahan and Weinberg wondered, as there were a variety of cancers, how many regulatory mechanisms were affected, or were the same regulatory mechanisms destroyed in the cell to become cancerous? Additionally, were there common signals? After asking these questions, they proposed that cancer cells exhibited six biological phenomena, which they called “hallmarks of cancer”. They proposed that these six hallmarks play a critical role in cancer development and are likely to occur in all cancers [3]. In the coming years, Hanahan and Weinberg added four more cancer features to the list of characteristics with the ongoing studies. Subsequently, with this update, the ten hallmarks of cancer approved today (eight trait characteristics and two enabling characteristics) were formed. These hallmarks include: (i) activating invasion and metastasis; (ii) enabling replicative immortality; (iii) evading growth suppressors; (iv) avoiding immune destruction; (v) genome instability and mutation; (vi) inducing angiogenesis; (vii) deregulating cellular energetics; (viii) resisting cell death; (ix) sustaining proliferative signaling; and (x) tumor-promoting inflammation [4]. Hanahan suggested, in his latest article, that potential features such as cellular plasticity, non-mutative epigenetic reprogramming, polymorphic microbiomes, and impaired differentiation could be added to the list in the future. However, he believes that the definition of these conceptual features needs to be discussed and experimentally validated with cancer biology studies [5].

Since the hallmarks of cancer genes and proteins orchestrate and drive carcinogenesis to a large extent, it is noteworthy to study these features in different cancers to understand the process of tumorigenesis and discover measurable indicators of altered biological states such as biomarkers. A study which was conducted by Nagy et al. [6] performed a pan-cancer survival analysis of cancer hallmark genes by not considering the associations among them. However, the development of cancer is generally induced by a combination of genes, proteins, metabolites, and other factors. In 2019, Yu and co-workers, by considering the association between genes, identified cancer hallmarks based on the gene co-expression networks for seven cancers [7]. Nevertheless, deciphering the changes at the protein level (protein interactome) is essential to understand tumorigenesis at the systemic level. Therefore, deciphering the hallmarks of different cancers using protein–protein interactions (PPIs) within the protein interactome is a promising approach to understand the mechanisms of cancer and to propose diagnostic, prognostic or therapeutic biomarker targets.

In our study, we performed a pan-cancer analysis and aimed to map differential hallmarks of cancer-associated protein–protein interactions (Figure 1). We examined the Cancer Genome Atlas (TCGA) transcriptome data from 12 different cancers: breast invasive carcinoma (BRCA); colon adenocarcinoma (COAD); head and neck squamous cell carcinoma (HNSC); kidney renal clear cell carcinoma (KIRC); kidney renal papillary cell carcinoma (KIRP); liver hepatocellular carcinoma (LIHC); lung adenocarcinoma (LUAD); lung squamous cell carcinoma (LUSC); prostate adenocarcinoma (PRAD); stomach adenocarcinoma (STAD); thyroid carcinoma (THCA); and uterine corpus endometrial carcinoma (UCEC), having sufficient samples in both tumor and control groups (*n* > 30) and applied a differential protein interactome algorithm [8], to determine differentially interacting hallmark of cancer proteins (DIHCPs) and DIHCP-centered modules (i.e., DIHCPs and their interacting partners) that represent significant changes in their interaction patterns between the tumor and control phenotypes. The diagnostic and prognostic capabilities of the identified DIHCP-centered modules were assessed in order to identify the modules’ potential ability to function as system biomarkers. In addition, the druggability of the prognostic and diagnostic DIHCPs were investigated. Ultimately, this study presents candidate system biomarkers that may be useful for understanding tumorigenesis, developing novel diagnostic tools, and improving personalized treatment strategies for various cancers, with a focus on the hallmarks of cancer.

## 2. Materials and Methods

### 2.1. Collecting of Gene Expression Data

RNA-sequencing (RNA-seq) fragments per kilobase of transcript per million fragments mapped (FPKM) normalized gene expression data were collected from the Cancer Genome Atlas (TCGA) [9]. For this study, gene expression data of 12 cancers with at least 30 normal and tumor samples of 33 different cancers were collected from the TCGA. In total, gene expression data were collected from 6239 tumor and 637 matched normal tissue samples. The 12 cancer types studied and their sample numbers are represented in Figure 2.

### 2.2. Collection of Cancer Hallmark Genes

Genes representing all the ten hallmarks of cancer [4] were collected from two publicly available biological repositories which are Cancer Hallmark Genes (CHG) da-tabase [10] and the Catalogue of Somatic Mutations in Cancer (COSMIC) database [11] (v.95). A total of 1906 different cancer hallmark genes were collected from these two repositories. The GeneCards: The Human Gene Database [12] was used to define proteins encoded by the cancer hallmark genes.

### 2.3. Collection of Human Protein Interactome Data

Human PPI data were extracted from the BioGRID database [13] (MV-Physical-4.2.191), which contains 51,745 physical and experimentally detected PPIs among 10,177 human proteins. Integration of the obtained PPI interaction data with proteins encoded by genes for which gene expression data are available in TCGA and with proteins encoded by cancer hallmark genes resulted in a network consisting of 7422 PPIs among 1906 proteins.

### 2.4. Identification of Differentially Interacting Hallmark of Cancer Proteins

The differential protein interactome algorithm [8] was applied to the gene expression profiles of all 12 cancer types using the R platform [14] (version 4.0.2). Briefly, differential hallmarks of cancer-associated PPIs (dHCPPIs) were determined to detect changes in PPI patterns between tumors and controls. To this end, the algorithm uses gene expression profiling to predict the relative frequency of observation (q-value) for each PPI to represent the dHCPPIs.

The following criteria were chosen to determine dHCPPIs: (i) if the predicted q value for PPI is less than 0.10, the interaction is significantly suppressed in the tumor state; (ii) if the predicted q value for PPI is greater than 0.90, the interaction is significantly activated in the tumor state; (iii) the normalized frequency of observation in the tumor or normal phenotype is greater than 20%.

Application of the differential protein interactome algorithm yielded DIHCPs representing significant changes in their interaction patterns during the transition between the normal and tumor phenotypes. The DIHCPs were classified into 2 groups according to their interaction partners: (i) proteins with suppressed interactions in the tumor state and (ii) proteins with activated interactions in the tumor state. DIHCPs (the hub hallmark of cancer proteins) together with their interacting protein partners were designated as DIHCP-centered modules (modular structures around hubs). Further analyses were performed using the DIHCP-centered modules with at least 10 proteins. The dHCPPIs and DIHCP-centered modules were visualized using Cytoscape (v3.5.0) [15].

### 2.5. Diagnostic Performance Analyses of Differentially Interacting Hallmark of Cancer Protein Modules

Principal component analyses (PCA) were performed based on TCGA-derived gene expression profiles of the genes encoding DIHCPs in each DIHCP-centered module. Each simulation was performed with at least 30 randomly selected normal and 30 tumor samples, and the first 3 principal components representing the highest variances (at least 80% of the total variance) were considered in determining the sensitivity (the proportion of positive test results among diseased individuals) and specificity (the proportion of negative test results among healthy individuals) metrics. Simulations were repeated until the robustness of the average value of the sensitivity and specificity metrics was ensured. The DIHCP-centered modules with at least 90% of the sensitivity and specificity values were considered statistically significant in this study and were accepted as a diagnostic DIHCP-centered module.

### 2.6. Prognostic Performance Analyses of Differentially Interacting Hallmarks of Cancer Protein Modules

To evaluate the prognostic performance of the DIHCP-centered modules, clinical information on 12 cancer types was collected from TCGA and used in the prognostic performance analyses. Prognostic abilities were evaluated by Kaplan–Meier plots and the log-rank test. All analyses were performed using the Survival package in R [14] (version 4.0.2). Samples were analyzed according to the prognostic index (PI), which is the linear component of the Cox model (PI = β1 × 1 + β2x2 +… + βpxp, where xi is the expression value of each gene, βi is the coefficient obtained from the Cox fit). The hazard ratio (HR = (O1/E1)/(O2/E2)) was calculated using the ratio between the relative mortality rate in group 1 and the relative mortality rate in group 2, where O and E are the observed and expected number of deaths, respectively. DIHCP-centered modules with a log-rank *p*-value < 0.01 were considered as statistically significant and accepted as a prognostic DIHCP-centered module in this study.

### 2.7. Enrichment Analyses of Diagnostic and Prognostic Modules

To obtain clues about the biological characteristics of the diagnostic and prognostic modules, over-representation analyses were performed using the bioinformatics tool Database for Annotation, Visualization and Integrated Discovery (DAVID) [16] to identify functional annotations (i.e., biological pathways) significantly associated with DIHCPs. Pathway p-values were determined using Fisher’s exact test, and the Benjamini–Hochberg correction was used as a correction technique for multiple testing. Pathways with adjusted *p* < 0.01 were considered statistically significant.

Moreover, although the DIHCPs were already associated with hallmarks of cancer and to determine which cancer hallmarks are prominent and to further examine the distribution of the hallmarks, the components of the DIHCP-centered modules (i.e., DIHCPs) were subjected to enrichment analysis. For the analyses, we used the obtained cancer hallmark genes from the two repositories [10,11] and integrated them with the diagnostic and prognostic DIHCPs.

### 2.8. Druggability of the Diagnostic and Prognostic Differentially Interacting Hallmark of Cancer Protein Modules

The DIHCP-centered modules that were (i) accepted as diagnostic in this study, (ii) accepted as prognostic in this study, and (iii) all dHCPPIs that were suppressed or activated in the tumor state were included in the druggability analysis. Components of the DIHCP-centered modules that met the established criteria were screened for druggability using the Drug Gene Interaction Database (DGIdb v4.2.0) [17]. Only FDA-approved drugs were considered throughout the screening process. The types of protein–drug interaction (activator, suppressor, inhibitor, etc.) were also considered. Namely, if all dHCPPIs in the modules were suppressed, the drug candidates with activator activity were considered, or if all dHCPPIs in the modules were activated, the drug candidates with inhibitor activity were considered.

## 3. Results

### 3.1. Interpreting Differential Protein Interactome Algorithm in Human Cancers

To identify dHCPPIs between tumor and normal tissue samples, the differential protein interaction algorithm [8] was independently applied to the gene expression profiles of the 12 cancer types and integrated with the reconstructed human protein interactome (7422 PPIs between 1906 proteins). Application of the algorithm yielded the hub proteins representing significant changes in the interaction patterns between “tumor phenotype” and “normal phenotype”, which we named “DIHCPs”.

We identified a total of 4405 dHCPPIs among 832 different hallmarks of cancer proteins (i.e., DIHCPs) for 12 cancer types. While HNSC had the highest number of dHCPPIs, KIRP had the lowest dHCPPIs. The tumor specificity of dHCPPIs varied by cancer type. Of all the dHCPPIs found, 786 (18.5%) of the dHCPPIs were specific to a cancer type. COAD had the highest number of specific interactions (230 specific dHCPPIs), whereas there was one specific dHCPPI for LUAD (0.2%) (Figure 3A). In addition, HNSC had the highest number of nonspecific dHCPPIs. On the other hand, none of the dHCPPIs was the same for all cancer types studied, and five interactions were the same in at most seven different cancers (i.e., CAV1-CTNNB1, COL1A1-IGFBP3, LDHA-LDHB, PCNA-GAPDH, and PSMB7-PSMB3). Comparative analysis of the interaction type (i.e., activated or suppressed) of these dHCPPIs revealed that the dHCPPIs were generally activated in the tumor state (88% activated).

### 3.2. Differentially Interacting Hallmark Proteins and Modules in Human Cancers

Of the 832 DIHCPs identified, 26% of the DIHCPs were specific for one cancer type. There were two common proteins (EGFR and ESR1) that had DIHCP features in all cancers studied. COAD had the highest number of specific DIHCPs. Namely, 16.9% of the DIHCPs of COAD were COAD-specific (Figure 3B).

The differential interactome showed a network topology with modular organization. Therefore, the DIHCPs (hub cancer hallmark proteins) together with their interacting protein partners, were referred to as modules, which we called “DIHCP-centered modules”. The DIHCP-centered modules with at least 10 components were considered, and each DIHCP-centered module was named after the name of the hub protein of the module.

In the comparative analysis of DIHCP-centered modules, we found that modules whose hubs belong to the protease or 14-3-3 protein complex shared the vast majority of DIHCPs. Therefore, DIHCP-centered modules sharing at least 70% of the common proteins were pooled based on cancer type to increase the robustness of DIHCP-centered modules. These pooled modules were designated as mPSMcomp or mYWHAcomp. For BRCA, four proteasome complex-associated hub modules were pooled. For COAD, two proteasome and two 14-3-3 protein complex-associated hub modules were pooled separately. In HNSC, 25 proteasome-associated and 3 of the 14-3-3 protein complex-associated hub modules were pooled. In LUAD and UCEC, five protease complex-associated hub modules were pooled independently. In LUSC, 31 proteasome complex-associated hub modules were pooled.

As a result, a total of 111 DIHCP-centered modules were identified (Appendix A). HNSC provided the highest number of DIHCP-centered modules (21 DIHCP-centered modules), while KIRP provided only 2 DIHCP-centered modules (Figure 4).

### 3.3. Diagnostic and Prognostic Power of Differentially Interacting Hallmarks of Cancer Protein-Centered Modules

In this study, the DIHCP-centered modules have the ability to be potential systems’ biomarkers because we believe that the potential disease differences depending on diseased and control status are mainly due to the coordinated action of a group of biological entities. We further hypothesize that the DIHCP-centered module, as a systems’ biomarker, would have high diagnostic and prognostic capabilities.

The diagnostic feature of each module was analyzed by PCA. Considering the most significant principal components (accounting for at least 80% of the total variance), sensitivity and specificity metrics were calculated, and modules with at least 90% of the sensitivity and specificity values were considered as diagnostic DIHCP-centered modules. Of the 111 DIHCP-centered modules, 39 had significantly high diagnostic performance (sensitivity ≥ 90% and specificity ≥ 90%), and some of these are shown in Figure 5. No diagnostic DIHCP-centered module could be observed for PRAD and THCA (Appendix A).

The prognostic performance of the modules was assessed using Kaplan–Meier survival plots. The log-rank p-value and hazard ratios were considered to determine whether the DIHCP-centered modules had a high impact on overall patient survival. A total of 88 DIHCP-centered modules had a high impact on patients’ overall survival (log-rank *p*-value < 0.01) in different cancer types (Figure 6), except for PRAD that had no effect on the prognostic DIHCP-centered modules (Appendix A).

Among the DIHCP-centered modules, there were a total of 30 DIHCP-centered modules that showed a high diagnostic or prognostic performance in any of the cancer types: one module in BRCA (mPPP1CA), KIRC (mCCND1) and STAD (mHSP90AA1); two modules in HNSC (mCAV1 and mEGFR), KIRP (mHSPB1 and mMET) and LIHC (mHSP90AA1 and mHSPB1; three modules in UCEC (mGNB1, mPCNA and mSFN); four modules in LUAD (mNPM1, mPPP1CA, mSTAT1, and mVIM); seven modules in COAD (mCDC37, mCTNNB1, mGNB2, mJUN, mMYC, mNONO, and mYWHAcomp); and LUSC (mCAV1, mCDH1, mNONO, mPCNA, mPPP1CA, mPSMcomp, and mYWHAG) indicated both diagnostic and prognostic properties.

### 3.4. Enrichment Analyses of Diagnostic and Prognostic Protein Modules

Pathway and hallmark enrichment analyses were performed to obtain further biological characteristics of the diagnostic and prognostic 30 modules. The DIHCPs pathway over-representation analysis of modules based on annotations stored in the KEGG database revealed various pathways (Figure 7). For instance, cancer-associated pathways such as bladder cancer, chronic myeloid leukemia, gastric cancer, glioma, and melanoma come into prominence. Signaling pathways such as Hippo, JAK-STAT, MAPK, PI3K-AKT, and WNT, which are known to be associated with cancer development and progression, were found to be statistically significant. Interestingly, viral carcinogenesis and viral infections associated with cancer (i.e., HBV, HPV, HTLV1, and KSHV) [18] were remarkable pathways associated with DIHCPs. Moreover, the hallmarks of cancer comprise ten distinct biological capabilities derived during the multi-staged development of tumors; however, to reveal the most fundamental trait of cancer cells, hallmark enrichment analyses were performed in diagnostic and prognostic DIHCP-centered modules (Figure 8A). Among the ten hallmarks of cancer, DIHCPs were most enriched in sustaining proliferative signaling, presumably due to the fact that unlimited replication is essential in cancer cells for tumor development. In addition, genome instability and mutation and enabling replicative immortality hallmarks remain in the background when compared to others.

### 3.5. Druggability of Differentially Interacting Hallmarks of Cancer Proteins

The DIHCPs were promising therapeutic targets because they represent significant changes in the interaction patterns between tumor and normal phenotype. For this reason, we investigated the druggability of the modules. Of the 30 diagnostic and prognostic DIHCP-centered modules, the DIHCPs of 17 modules were examined according to the established third criteria, because 13 of the dHCPPIs modules had both suppressed and activated interactions that did not meet our third criteria (see Materials and Methods).

We found that of the 17 DIHCP-centered modules, 28% of the DIHCPs were druggable. The mPSMcomp belonging to LUSC had the highest percentage (66%) of druggable proteins, whereas the lowest number of druggable DIHCPs among all cancers analyzed belonged to mGNB2 (10%) of COAD (Figure 8B).

A total of 200 different drugs targeting DIHCPs were found. A total of 54% of the drugs were specific for only one cancer type. None of the drugs were specific for all cancers studied, and 25 of the drugs were specific for no more than 6 different cancers (Appendix A).

## 4. Discussion

Complex diseases such as cancer usually arise as a result of interactions between biological entities. The biological molecules that trigger the development of a cancer or biological process never function alone; they work together in a complex, interconnected network to carry out these biological functions. The state of interconnectedness of molecules establishes the phenomenon of the “biological network”. A biological network is an advanced concept that allows researchers to model, characterize, and decipher complex interactions between different biologically relevant entities in a biological system [19]. Biological networks thus change our perspective and shed light on the internal organization of the cell, allowing us to understand the mechanisms underlying complex diseases. Moreover, predictive, preventive, and personalized medicine is a holistic healthcare strategy that aims to predict individual predisposition, provide targeted prevention, and provide personalized treatment [20]. To advance personalized medicine strategies, the discovery of new treatments is essential. However, with today’s financial resources, it is nearly impossible to discover drugs from scratch using traditional methods. Using biological network analysis, researchers can also discover and/or predict anti-cancer drugs [21]. We believe that network-based systems biology is a new approach for discovering treatment and prevention strategies, especially for cancer, which is currently a major burden worldwide, and researchers are embracing this concept. For example, researchers have integrated gene expression profiling with multiple networks to identify new biomarkers and drug candidates for breast cancer [22], pancreatic cancer [23], cervical cancer [24], acute myeloid leukemia [25], thyroid cancer [26], and so on. Accordingly, these holistic analyses enable the discovery of potential biomarkers for diagnosis, prognosis, or therapeutic purposes. Therefore, biological networks hold great potential for the present and the future, especially for complex diseases such as cancer [27].

The protein–protein interactome represents a biological network that encompasses all physical protein interactions within the cell. These protein–protein interactions play a crucial role in the living organism. These are essential for almost all cellular functions, involved in the physiology and biochemistry of the organism. These can also influence cancer cell growth, transmit oncogenic signals, and even cause the development of typical cancer features. Targeting such interactions is a promising strategy for new drug development [28,29]. Thus, it is strongly suggested that significant changes in protein–protein interactions occur depending on the phenotype of the individual (i.e., diseased versus healthy) [30].

We applied our constructed differential interactome algorithm, which had proven to be a powerful tool in other studies [8,31,32], for the protein–protein interactome data on hallmarks of cancer. Our results revealed DIHCPs showing remarkable changes in the interaction patterns of patients during the transition from the “normal” phenotype to the “cancer” phenotype. As mentioned earlier, cancers occur and spread usually due to the coordinated action of a group of biological molecules. In this study, DIHCPs and the modular structures around these proteins (DIHCP-centered modules) were considered as potential systems’ biomarkers. However, we believe that a valid biomarker must provide prognostic or diagnostic information about the specific cancer. For this reason, to provide confidence in the precise diagnostic and prognostic capacity of the systems’ biomarkers we clearly demonstrated their diagnostic and prognostic performances by PCA and survival analyses and demonstrated high biomarker performances in multiple tumors (30 diagnostic and prognostic DIHCP-centered modules).

In this study, we only used hallmark of cancer proteins, while we integrated proteomics data to apply the differential interactome algorithm. Hallmarks of cancer proteins are considered as driver proteins of tumorigenesis. These proteins are responsible for the most basic phenotypic features of tumor initiation and progression [3]. Therefore, we hypothesized that we can highlight the principles and mechanisms of tumorigenesis by focusing only on the proteins essential for carcinogenesis (hallmarks of cancer proteins), we can reduce the complexity of the disease, and typify the process by characteristic and complementary features. In addition, if the potential system biomarker components (i.e., DIHCPs) presented in the study are all among the hallmark characteristics of human cancers, we believe we will identify highly robust cancer biomarkers. Moreover, since these hallmarks represent features of cancer cells, recent therapeutic approaches have been increasingly aiming at targeting hallmark proteins [33]. Therefore, the systems’ biomarkers identified in this study will have remarkable clinical value. For this reason, this study also evaluated the druggability of the diagnostic and prognostic systems’ biomarkers.

The hubs of the diagnostic and prognostic systems’ biomarkers are mostly associated with the sustaining proliferative signaling involved as an important feature in the hallmarks of cancer. It is known that cancer cells can stimulate their own proliferation indefinitely and this cancer hallmark characteristic is chronically activated in all cancers. The well-known mechanism for sustaining proliferative signaling is to activate oncogenes [34]. Oncogenes encode proteins that stimulate cell proliferation and programmed cell death. An oncogene arises when a proto-oncogene is altered genetically (i.e., mutated) and over-expressed. For instance, in cancers, the most noticeable oncogenes are BRAF and RAS [35]. Moreover, resisting the cell-death hallmark feature comes into prominence for diagnostic and prognostic hubs. Resistance to cell death is a natural barrier to cancer development and is mostly regulated by programmed cell death. The best-known form of programmed cell death is apoptosis, in which cells are destined to die. In addition to apoptosis, necroptosis is referred to as the second important process for inducing cell death. Apoptosis and necroptosis cause different immune responses. Necroptosis releases molecules that promote inflammation, while apoptosis triggers silent immunological responses [36]. Autophagy is another process that leads to cell death [34]. The autophagy process allows cells to remove unnecessary or dysfunctional components. In cancer, autophagy may play a tumor-suppressor or oncogene role under certain conditions and at certain stages of carcinogenesis [37]. A recent review systematically described the hallmarks of different cell death modes [38].

The major limitation of the study is the lack of experimental validations of the diagnostic and prognostic modules with relevant tissue samples or cell lines. Therefore, the most important aspect of this study is to translate these computational findings into experimental approaches. For example, future in vitro studies need to be performed to investigate the impact of the identified modules in terms of their response to cell viability, cell migration and disease progression. In addition, the accuracy, consistency, reproducibility, and reliability of the biomarkers presented in this study should be experimentally validated if they prove to be clinically useful. In addition, the mechanism of action of these modules needs to be experimentally evaluated to clearly elucidate their effects on the hallmarks of cancer characteristics. We believe that computational analysis is an important and first step in biomarker development. However, to address the broad medical and scientific audience, the need for experimental validation is inevitable.

In conclusion, the knowledge of specific proteins being impacted within a given hallmark of the patient’s cancer type could offer the opportunity to personalize treatments. One such approach would be to target proteins from the proteins in the enrichment analysis that match the patient’s specific tumor type and prioritize repurposing the drugs proposed from our analysis. Furthermore, our investigation on the hallmarks of cancer protein provides valuable data for further experimental and clinical efforts in a variety of cancer types since the proposed systems’ biomarkers have the potential to be diagnostic and/or prognostic. Moreover, the protein–protein interactions could be utilized as therapeutic targets.

## Figures and Tables

**Figure 1 jpm-12-01919-f001:**
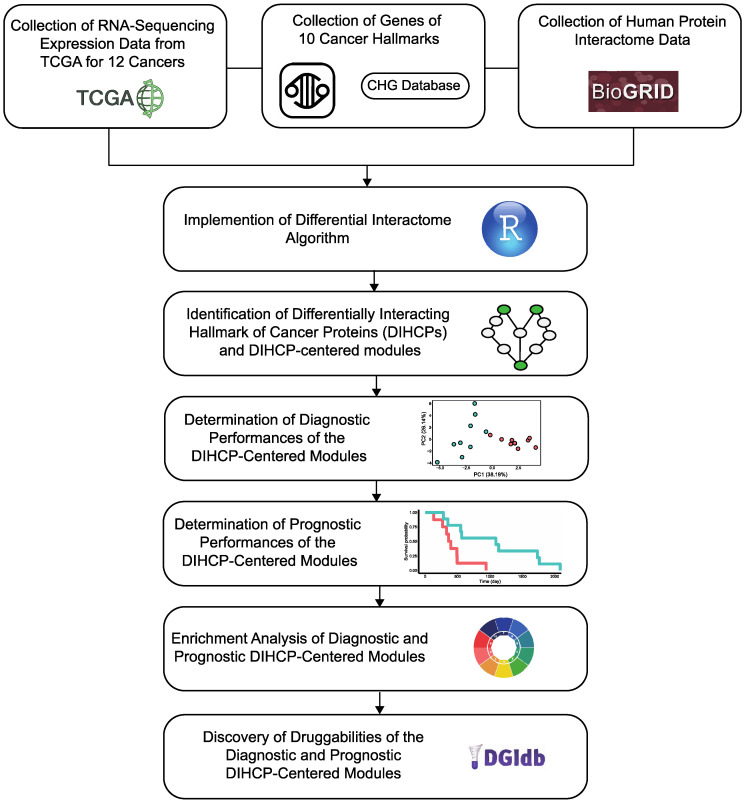
The systematic approach for the study. The steps of the applied systems’ biology methodology for the study. After collecting the data from different repositories, the differential interactome algorithm was applied. This algorithm was used to identify differentially interacting hallmark of cancer proteins (DIHCPs) and DIHCPs-centered modules. The prognostic and diagnostic performance of the DIHCPs-centered modules was evaluated. The enrichment analyses were performed and finally the components of the modules’ druggability were discovered.

**Figure 2 jpm-12-01919-f002:**
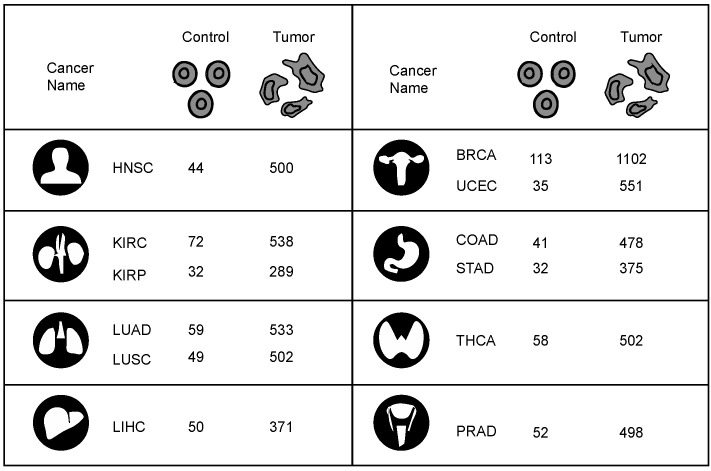
Distribution of samples from 12 human cancers. The diagram shows control and tumor samples from 12 human cancers analyzed in the study.

**Figure 3 jpm-12-01919-f003:**
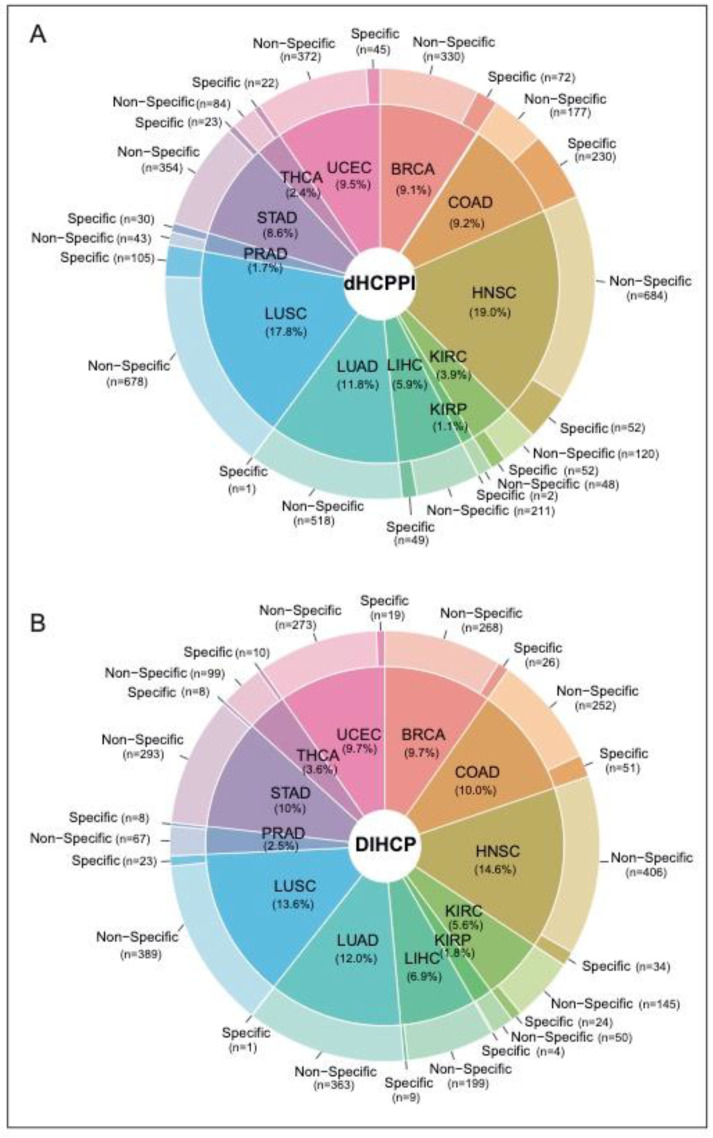
Distribution of differential hallmark of cancer-associated protein–protein interactions and differentially interacting hallmark of cancer proteins. (**A**) The pie-donut chart showing the percentage of specific and non-specific differential hallmark of cancer-associated protein–protein interaction (dHCPPIs) between cancer types. (**B**) The pie-donut chart showing the percentage of specific and non-specific differentially interacting hallmark of cancer proteins (DIHCPs) between cancer types.

**Figure 4 jpm-12-01919-f004:**
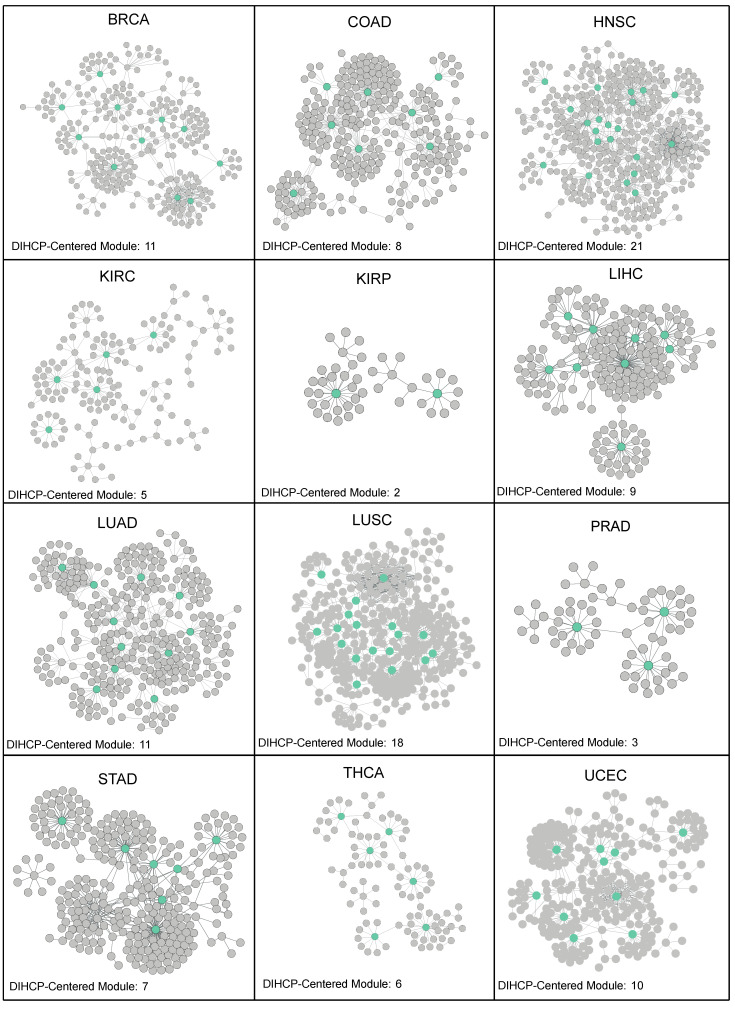
Differential interactome networks in 12 human cancers. For each cancer type, a differential interactome network was constructed around a differential hallmark of cancer-associated PPIs (dHCPPIs). The number of differentially interacting hallmarks of cancer proteins (DIHCPs) for each cancer type was presented.

**Figure 5 jpm-12-01919-f005:**
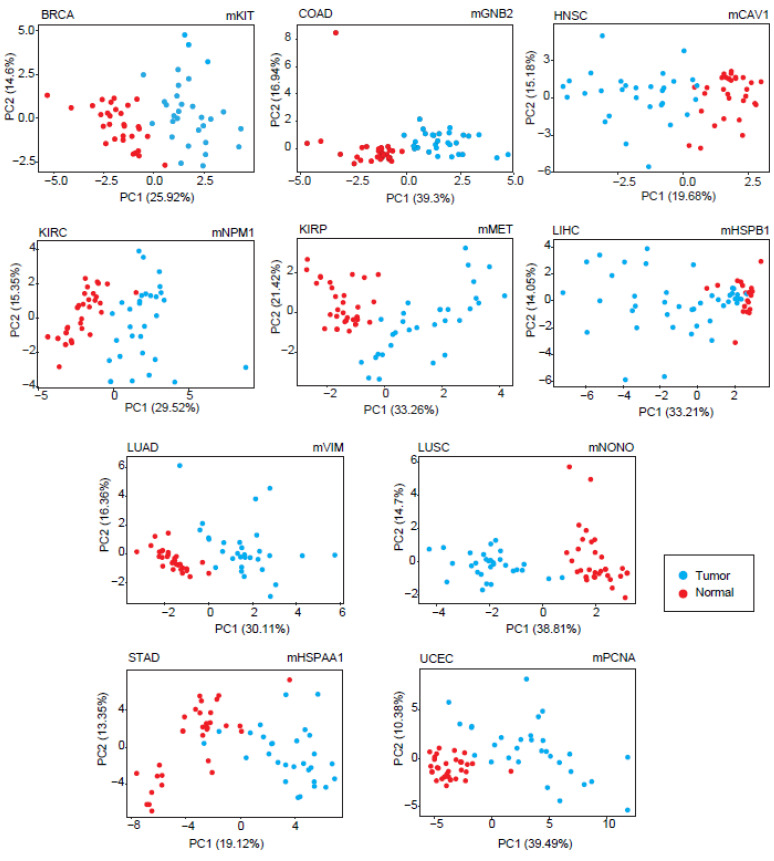
Principal component analyses for different cancer types. PCA plots showing individual differences in protein expression profiles between cancer types comprising at least 30 individuals in each type.

**Figure 6 jpm-12-01919-f006:**
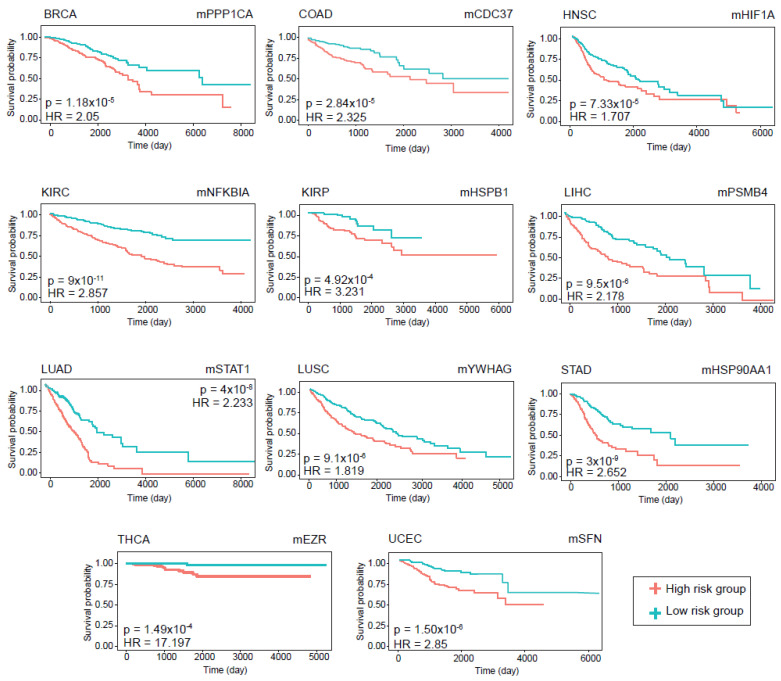
Prognostic analyses for different types of cancer. Kaplan–Meier Plots estimating patients’ survival for 11 cancers with p-value and hazard ratio given for each curve.

**Figure 7 jpm-12-01919-f007:**
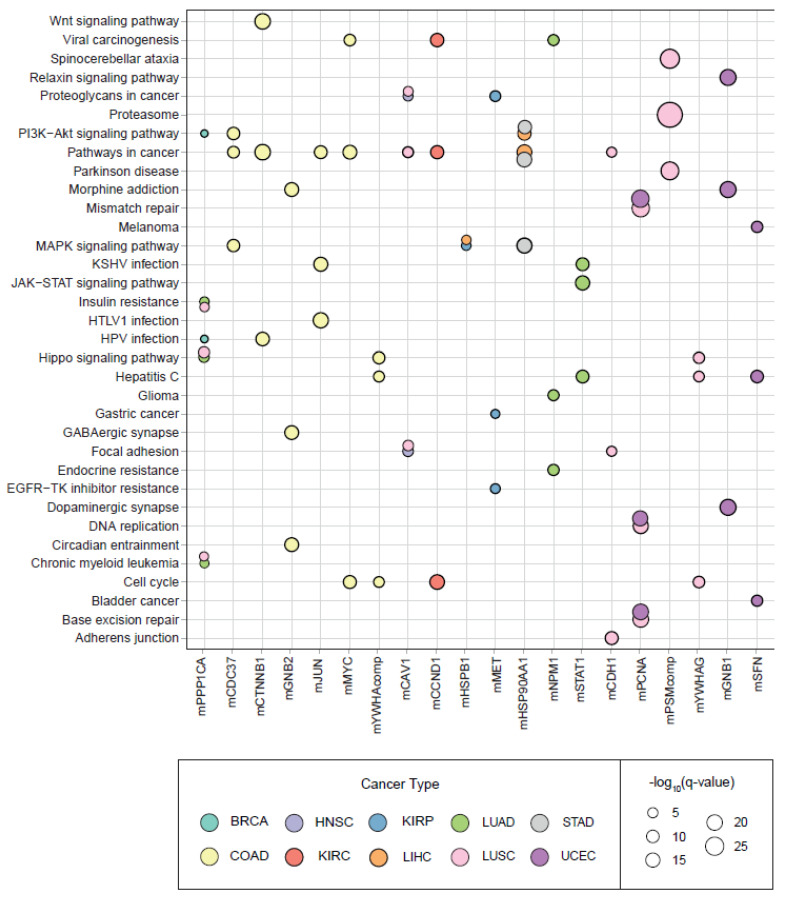
The pathway enrichment analyses results. The bubble plot indicates the diagnostic and prognostic 30 modules differentially interacting hallmark of cancer protein’s (DIHCP’s) pathway enrichment results according to q-value (−log_10_) significance.

**Figure 8 jpm-12-01919-f008:**
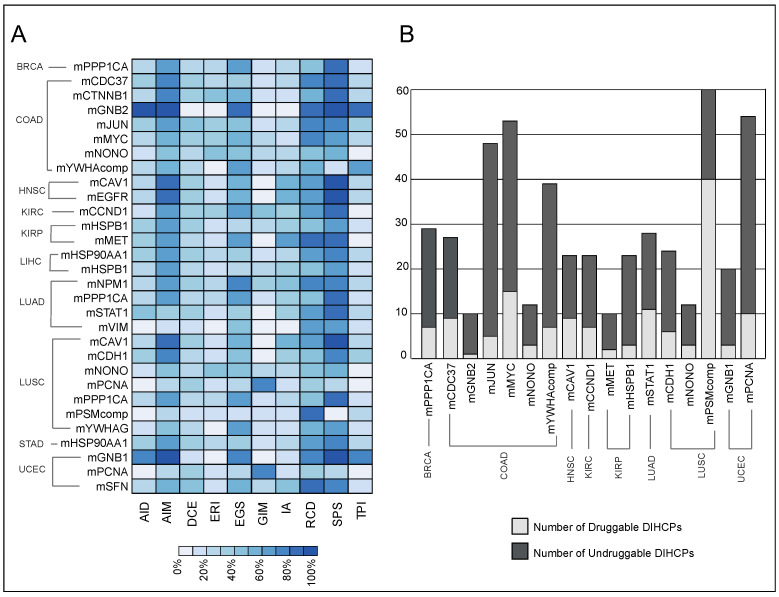
The hallmark enrichment and druggability analyses results. (**A**) The heat-map represents the hallmark enrichment results of the module’s differentially interacting hallmark of cancer proteins (DIHCPs). AID: avoiding immune destruction; AIM: activating invasion and metastasis; DCE: deregulating cellular energetics; ERI: enabling replicative immortality; EGS: evading growth suppressors; GIM: genome instability and mutation; IA: inducing angiogenesis; RCD: resisting cell death; SPS: sustaining proliferative signaling; TPI: tumor-promoting inflammation. (**B**) The druggability distribution of module’s DIHCPs. The bar graph indicates the number of DIHCPs, which are druggable (light grey bars) and undruggable (dark grey bars).

## Data Availability

Publicly available datasets were analyzed in this study. The datasets analyzed during the current study are available in The Genome Cancer Atlas (https://portal.gdc.cancer.gov/, accessed on 1 March 2022). Protein interactome data are available in Biological General Repository for Interaction Datasets (https://thebiogrid.org, accessed on 30 March 2022). The cancer hallmark gene data are available in the Catalogue of Somatic Mutations in Cancer (https://cancer.sanger.ac.uk/cosmic, accessed on 23 May 2022) and Cancer Hallmark Genes (http://www.bio-bigdata.com/CHG/index.html, accessed on 23 May 2022). Source codes for the differential interactome algorithm (implemented in R, version 4.0.2) are freely available at http://sysbio.bioe.eng.marmara.edu.tr/diff-int-ome, accessed on 25 July 2022.

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
