# Peer review of "A Pan-Cancer Atlas of Differentially Interacting Hallmarks of Cancer Proteins"

_jpm, 2022, doi:10.3390/jpm12111919_

Round 1

Reviewer 1 Report

In the manuscript entitled “A Pan-Cancer Atlas of Differentially Interacting Hallmarks of Cancer Proteins”, authors have performed a pan-cancer analysis and mapped differential hall-marks of cancer-associated protein-protein interactions. Doing the same, they have examined the Cancer Genome Atlas (TCGA) transcriptome data from 12 different cancers and applied differential protein interactome algorithm. The diagnostic and prognostic capabilities of the identified DIHCP-cantered modules were also assessed in order to identify the modules potential ability to function as system biomarkers. In addition, the druggabilities of the prognostic and diagnostic DIHCPs were also investigated. This overall study has presented candidate system biomarkers which may be used to understand tumorigenesis, development of novel diagnostic tools, and improving personalized treatment strategies for various cancers. Besides all, authors need to improve the manuscript in respect of following points:

There are some minor concerns that I would like the authors to address.

(i)               I guess the figures are designed and exported in Microsoft Office tools, please increase the export resolution to have more clear understanding of the diagrammatic representation.

(ii)             Please provide future aspects of this studies.

(iii)           Have similar studies been done, and if so, how well do they compare with earlier studies? 

(iv)           How reliable are the predicted results?

(v)              The reference given are quite old and also there are some errors in the year of publication, author must include some recent references. Please cite recent articles related to the studies like  "Irinotecan and vandetanib create synergies for treatment of pancreatic cancer patients with concomitant TP53 and KRAS mutations." Briefings in bioinformatics 22, no. 3 (2021): bbaa149; "Bioinformatics Approaches for Anti-cancer Drug Discovery". Current drug targets, 2020. 21(1): p. 3-17; "An integrated systems biology and network-based approaches to identify novel biomarkers in breast cancer cell lines using gene expression data." Interdisciplinary Sciences: Computational Life Sciences (2020): 1-14.

(vi)           The manuscript has to be read by some native English speaker so that more readers can take benefit of it.

This is a good and significant article. The work presented by the authors is of high scientific interest. I strongly believe the outcomes are worthy enough to be published. I would like to see the revised version of this manuscript.

Author Response

Response to REVIEWER-1

(i) I guess the figures are designed and exported in Microsoft Office tools, please increase the export resolution to have more clear understanding of the diagrammatic representation.

Thank you for your suggestion. The figures were created using the R programming language and the Adobe Illustrator graphics program. The illustrations were submitted to the journal's submission system in PDF format. These files contain vector graphics and are not resolution sensitive. We assumed this was due to the submission system. However, at your suggestion, we have revised and enlarged the figures to make them clearer for readers.

(ii) Please provide future aspects of this studies.

Thank you for your suggestion, we have included the future aspect of our study in the revised manuscript by adding the following paragraph to the Discussion section:

The major limitation of the study is the lack of experimental validations of the diagnostic and prognostic modules with relevant tissue samples or cell lines. Therefore, the most important aspect of this study is to translate these computational findings into experimental approaches. For example, future in vitro studies need to be performed to investigate the impact of the identified modules in terms of their response to cell viability, cell migration and disease progression. In addition, the accuracy, consistency, reproducibility, and reliability of the biomarkers presented in this study should be experimentally validated if they prove to be clinically useful. In addition, the mechanism of action of these modules needs to be experimentally evaluated to clearly elucidate their effects on the hallmarks of cancer characteristics. We believe that computational analysis is an important and first step in biomarker development. However, to address the broad medical and scientific audience, the need for experimental validation is inevitable.

 (iii) Have similar studies been done, and if so, how well do they compare with earlier studies?

Actually, the applied differential protein interactome algorithm was developed by our research group and as mentioned in the manuscript, this algorithm was applied to different types of cancer. However, in this study, we emphasize the cancer hallmarks during our analyzes, which was implemented for the first time. Unlike other works, we presented diagnostic and prognostic biomarkers consisting only of established cancer hallmark proteins. Since hallmarks of cancer proteins are considered to be driver proteins of tumorigenesis, we believe that the presented biomarkers are more likely to be very robust cancer biomarkers and have remarkable clinical value.

(iv) How reliable are the predicted results?

In recent decades, advanced methods have generated a large amount of data at various molecular levels and they are available as publicly accessible complex biological datasets. In parallel with the accumulation of these data, a biological phenomenon known as "systems biology" has emerged. Systems biology approaches allow us to study a biological system as complex sets of binary interactions or relationships between different biomolecules inherent in the biological system. Although this is a computational study, prediction of the results is necessary. However, we should keep in mind that the prediction of unobserved variables or the prediction of something we have no idea about or predict from scratch cannot be estimated. Therefore, we believe that systems biology approaches provide researchers clues about what should be predicted experimentally. We believe that we have provided valuable data in this study and suggest to researchers what they should predict. However, we know that experimentally predicting these results requires a lot of time and a large budget. Therefore, we present our results to the scientific world and help researchers make experimental hypotheses.

(v) The reference given are quite old and also there are some errors in the year of publication, author must include some recent references. Please cite recent articles related to the studies like  "Irinotecan and vandetanib create synergies for treatment of pancreatic cancer patients with concomitant TP53 and KRAS mutations." Briefings in bioinformatics 22, no. 3 (2021): bbaa149; "Bioinformatics Approaches for Anti-cancer Drug Discovery". Current drug targets, 2020. 21(1): p. 3-17; "An integrated systems biology and network-based approaches to identify novel biomarkers in breast cancer cell lines using gene expression data." Interdisciplinary Sciences: Computational Life Sciences (2020): 1-14.

Thank you very much for your suggestion. According to your recommendation, we have added a short paragraph to the Discussion section and have reviewed and discussed the articles you mentioned. You can find the added paragraph and the relevant citations below.

“Moreover, predictive, preventive, and personalized medicine is a holistic healthcare strategy that aims to predict individual predisposition, provide targeted prevention, and provide personalized treatment [20]. To advance personalized medicine strategies, the discovery of new treatments is essential. However, with today's financial resources, it is nearly impossible to discover drugs from scratch using traditional methods. Using biological network analysis, researchers can also discover and/or predict anti-cancer drugs [21]. We believe that network-based systems biology is a new approach for discovering treatment and prevention strategies, especially for cancer, which is currently a major burden worldwide, and researchers are embracing this concept. For example, researchers have integrated gene expression profiling with multiple networks to identify new biomarkers and drug candidates for breast cancer [22], pancreatic cancer [23], cervical cancer [24], acute myeloid leukemia [25], thyroid cancer [26], and so on.

[20] Goetz, LH,; Schork, N.J. Personalized medicine: motivation, challenges, and progress. Fertil Steril. 2018, 109, 952-963.

[21] Li, K.; Du, Y.; Li, L.; Wei, D.Q. Bioinformatics Approaches for Anti-cancer Drug Discovery. Curr. Drug Targets. 2020, 21, 3–17.

[22] Khan, A.; Rehman, Z.; Hashmi, H. F.; Khan, A.A.; Junaid, M.; Sayaf, A. M.; Ali, S. S.; Hassan, F. U.; Heng, W.; Wei, D.Q. An Integrated Systems Biology and Network-Based Approaches to Identify Novel Biomarkers in Breast Cancer Cell Lines Using Gene Expression Data. Interdiscip Sci. 2020, 12(2):155–168.

[23] Kaushik, A. C.; Wang, Y.J.; Wang, X.; Wei, D.Q. Irinotecan and vandetanib create synergies for treatment of pancreatic cancer patients with concomitant TP53 and KRAS mutations. Brief Bioinform. 2021, 22, bbaa149.

[24] Kori, M.; Arga, K.Y.; Mardinoglu, A.; Turanli, B. Repositioning of Anti-Inflammatory Drugs for the Treatment of Cervical Cancer Sub-Types. Front Pharmacol. 2022, 13, 884548.

[25] Kelesoglu, N.; Kori, M.; Turanli, B.; Arga, K.Y.; Yilmaz, B.K.; & Duru, O.A. Acute Myeloid Leukemia: New Multiomics Molecular Signatures and Implications for Systems Medicine Diagnostics and Therapeutics Innovation. OMICS. 2022, 26, 392–403.

[26] Gulfidan, G.; Soylu, M.; Demirel, D.; Erdonmez, H.; Beklen, H.; Ozbek Sarica, P.; Arga, K.Y.; Turanli, B. Systems biomarkers for papillary thyroid cancer prognosis and treatment through multi-omics networks. Arch Biochem Biophys. 2022, 15, 109085.

(vi) The manuscript has to be read by some native English speaker so that more readers can take benefit of it.

Thank you for your suggestion. We have paid strict attention to language in the revision period.

Reviewer 2 Report

The article 'A pan-cancer atlas of differentially interacting hallmarks of cancer proteins’ submitted by Kori M et al., identifies differentially interacting hallmarks of cancer proteins (DIHCPs) and their interacting partners (DIHCP-centered modules), which show a significant change in their interacting patterns between tumors and control phenotypes. The authors have used transcriptome data (cancer genome atlas (TCGA)) of 12 different cancer types for this study. Furthermore, authors have performed diagnostic and prognostic performance of DIHCP-centered modules. Authors have emphasized their study provides a list of candidate system biomarkers that could be useful in understanding cancer progression and therapy planning. I have enlisted a few concerns here-

1.     The authors should address the font size in the images of Fig.2, as it is difficult to read due to the small font size.

2.     What is written on the x- and y-axis of the graphs of Fig.4 and Fig.5. Half of the data is unclear since we don’t know what is written in these graphs.

3.     The authors should explain the characteristics of the 30 modules found in the study in the results section. Just mentioning the name is not sufficient.

4.     The discussion section requires improvement as authors did not discuss anything related to their findings. I could not differentiate between the introduction and discussion sections.

Author Response

Response to REVIEWER-2

  1. The authors should address the font size in the images of Fig.2, as it is difficult to read due to the small font size.

Accordingly, we split Figure 2 because we could not make the figure's font larger due to page size standards. Therefore, we created a new figure (Figure 1) for the revised manuscript that includes information on the number of samples of the 12 cancer types studied and Figure 2 B and Figure 2C were presented in the revised manuscript as Figure 4A and Figure 4B, respectively.

  1. What is written on the x- and y-axis of the graphs of Fig.4 and Fig.5. Half of the data is unclear since we don’t know what is written in these graphs.

The figures were created using the R programming language and the Adobe Illustrator graphics program. All figures were submitted in PDF format to the journal's submission system. We suspect that this is due to the submission system. However, at your suggestion, we have revised and enlarged the figures to make them clearer for readers.

  1. The authors should explain the characteristics of the 30 modules found in the study in the results section. Just mentioning the name is not sufficient.

Thank you for your recommendation. Accordingly, to gain more insights of the revealed diagnostic and prognostic modules we implemented overrepresentation analysis to DIHCPs. Hereunder, diagnostic and prognostic modules, overrepresentation analyzes were performed using the bioinformatics tool Database for Annotation, Visualization and Integrated Discovery (DAVID) [16] to identify functional annotations (i.e., biological pathways) significantly associated with DIHCPs. Pathway p-values were determined using Fisher's exact test, and the Benjamini-Hochberg correction was used as a correction technique for multiple testing. Pathways with adjusted p < 0.01 were considered statistically significant.

The resulting significant pathways were shown in a figure (Figure 7) in the revised manuscript and explained in the Results section, following as you suggested; Pathway and hallmark enrichment analyzes were performed to obtain further biological characteristics of the diagnostic and prognostic 30 modules. The DIHCPs pathway overrepresentation analysis of modules based on annotations stored in the KEGG data-base revealed various pathways (Figure 7). For instance, cancer-associated pathways such as bladder cancer, chronic myeloid leukemia, gastric cancer, glioma, and melanoma come into prominence. Signaling pathways such as Hippo, JAK-STAT, MAPK, PI3K-AKT, and WNT, which are known to be associated with cancer development and progression, were found to be statistically significant. Interestingly, viral carcinogenesis and viral infections associated with cancer (i.e., HBV, HPV, HTLV1, and KSHV) [18] were remarkable pathways associated with DIHCPs.

[16] Sherman, B.T.; Hao, M.; Qiu, J.; Jiao, X.; Baseler, M.W.; Lane, H.C.; Imamichi, T.; Chang, W. DAVID: a web server for functional enrichment analysis and functional annotation of gene lists (2021 update). Nucleic Acids Res. 2022, 50, W216–21.

[18] Kori M,; Arga K.Y. Human oncogenic viruses: an overview of protein biomarkers in viral cancers and their potential use in clinics. Expert Rev Anticancer Ther. 2022, 1-14.

  1. The discussion section requires improvement as authors did not discuss anything related to their findings. I could not differentiate between the introduction and discussion sections. 

Thank you very much for your comment. We have improved our discussion section based on your and Referee #1's suggestions. We hope the revised Discussion section meets your expectations.

Round 2

Reviewer 2 Report

Authors have addressed my concerns.